# Dual Component Polymeric Epoxy-Polyaminoamide Based Zinc Phosphate Anticorrosive Formulation for 15CDV6 Steel

**Omar Dagdag [1]**, **Ghadir Hanbali [2]**, **Bayan Khalaf [2]**, **Shehdeh Jodeh [2,\***, **Ahmed El Harfi [1]** and **Abdelhadi Deghles [3]**

[1] Laboratory of Agroresources, Polymers and Process Engineering (LAPPE), Department of Chemistry, Faculty of Science, Ibn Tofail University, BP 133, 14000 Kenitra, Morocco
[2] Department of Chemistry, An-Najah National University, P.O. Box 7 Nablus, Palestine
[3] Scientific Research and Development, Al Istiqlal University, P.O. Box 10 Jericho, Palestine
\* Correspondence: sjodeh@hotmail.com; Tel.: +970-599590498

**Abstract:** The present research is focused on a formulation with two active components as an anticorrosive polymer coating for 15CDV6 steel. The dual component formulation (epoxy-zinc phosphate (ZP) coating) consists of a polymeric epoxy resin Bisphenol A diglycidyl ether (DGEBA) cured with a polyaminoamide as a first component and zinc phosphate ($Zn_3(PO_4)_2(H_2O)_4$) (ZP) added in 5% by weight as a second component. The anticorrosive performance of the epoxy-ZP coating was evaluated against the standard coating, which consists of only one component, the cured polymeric epoxy resin. The two polymer coatings were evaluated by electrochemical impedance spectroscopy (EIS). The surface morphology was of the two polymer coatings were characterized by scanning electron microscopy (SEM). The coated samples of 15CDV6 steel were tested in a harsh environment of corrosive electrolytes (3 wt % NaCl solution). Under these conditions, a very high impedance value was obtained for 15CDV6 steel coated with the epoxy-ZP coating. Even after exposure for a long period of time (5856 h), the performance was still acceptable, indicating that the epoxy-ZP coating is an excellent barrier. The standard epoxy coating provided an adequate corrosion protection performance for a short period of time, then the performance started to decline. The results were confirmed by surface characterization, a cross-sectional image obtained by optical microscopy for an epoxy-ZP coating applied on 15CDV6 steel exposed for 5856 h to a salt spray test showed that the coating is homogeneous and adheres well to the surface of the steel. So, the coating with a dual component could have great potential in marine applications as anticorrosive for steel.

**Keywords:** polymeric epoxy resin; polyaminoamide; zinc phosphate; polymer coating; steel and saline

## 1. Introduction

Corrosion of steel alloys is one of the most important safety and economic concerns for many industries [1,2]. Because of its excellent mechanical power and relatively low cost, 15CDV6 steel is one of the frequently used steel-based materials for numerous applications in several industries. However, it is highly susceptible to corrosion during several industrial processes where metallic components undergo corrosive dissolution by the aggressive saline environment [3,4].

Organic coatings are usually applied as thin films on the surface of metals, thus preventing corrosive agents from reaching the metal surface [5–8]. Despite all the advantages of organic coatings, some improvement is still needed, for instance their life-time is limited, after a certain period they tend to deteriorate and their performance, such as their barrier properties, declines [9]. Durability and life time of an organic coating are controlled by several factors, among these are the chemistry of the

coating, the cross-linking density in the coating, and the functional group present on the coating surface which affects the strength of adhesion to the metal surface. Among organic coatings, water-based epoxy coatings are the most attractive due to their unique mechanical properties, superior adhesion to substrate, good thermal stability, excellent corrosion resistance and chemical resistance. In addition, they can be available at a low cost [10,11] and are nontoxic. These attractive properties could be attributed to high the cross-linking density and to the functional groups such as amines and hydroxyl present in the polymer coating [12]. However, long term exposure to corrosive materials, cause partial diffusion of corrosive species into the polymer coating from scratches or inherent pores. Eventually the diffusing materials will reach the metal–coating interface. This causes the initiation of corrosion reactions and slow pealing of the coatings [13–15].

Using inorganic pigments such as zinc chromate as a corrosion inhibitor has become popular in recent years [16]. They tend to improve the adhesion and anticorrosion performance of organic coatings. A couple of published studies showed that some of the inorganic pigments dissolve in water. The dissolved part tends to undergo oxidation and precipitate as a passive film on the metal surface [17,18]. The passive film can block the active zones on the metal surface and reduce the rate of electrochemical processes [19].

However, the toxicity and the carcinogenicity of some of the reported inorganic pigment have limited their use [13,20,21]. In recent decades, alternative nontoxic pigments were developed such as, zinc phosphate (ZP). In addition, zinc phosphate pigment showed better performance as a corrosion inhibitive for steel. This was attributed to it is solubility in water.

In the first part of the present study, a polymeric based Bisphenol A diglycidyl ether (DGEBA) cured with a polyaminoamide was prepared and coated on the surface of 15CDV6 steel with zinc phosphate pigments.

In the second part, we have evaluated and tested in an aggressive marine environment. The anticorrosive performance of the two polymer coatings was monitored by electrochemical impedance spectroscopy (EIS) and confirmed by SEM.

## 2. Experimental

### 2.1. Materials and Methods

The formulations of the polymer coatings are given in Table 1.

**Table 1.** Composition of organic model coatings.

| Component (Liquid Epoxy) | Weight Percent (wt %) | Supplier |
|---|---|---|
| DGEBA (average molecular weight < 700) | 25 ≤ wt % < 50 | MAPAERO-Aerospace Coatings and Finishes, Pamiers CEDEX, France |
| Nitroethane | 25 ≤ wt % < 50 | MAPAERO-Aerospace Coatings and Finishes, Pamiers CEDEX, France |
| Triglycidyl ether of trimethylolpropane | 10 ≤ wt % < 25 | MAPAERO-Aerospace Coatings and Finishes, Pamiers CEDEX, France |
| Alkoxysilane | 0 ≤ wt % < 2.5 | MAPAERO-Aerospace Coatings and Finishes, Pamiers CEDEX, France |
| **Component (Liquid Hardener)** | **Weight Percent (wt %)** | **Supplier** |
| Polyaminoamide | 10 ≤ wt % < 25 | MAPAERO-Aerospace Coatings and Finishes, Pamiers CEDEX, France |
| Butane-2-ol | 25 ≤ wt % < 50 | MAPAERO-Aerospace Coatings and Finishes, Pamiers CEDEX, France |
| Zinc oxide | 0 ≤ wt % < 2.5 | MAPAERO-Aerospace Coatings and Finishes, Pamiers CEDEX, France |
| Demineralized water ($\varrho > 1$ Ω·cm and $\sigma < 1$ μS/cm) | – | – |
| Methyl ethyl ketone (MEK) (≥99%) | – | Sigma Aldrich, Saint Louis, MO, USA |
| Zinc phosphate (ZP) (99.99%) | 0 ≤ wt % < 5 | Sigma Aldrich, Saint Louis, MO, USA |

Samples of 15CDV6 steel were used in this study, the composition of the steel samples are given in Table 2.

**Table 2.** Elementary chemical composition in weight percent (wt %) of the 15CDV6 steel that was used.

| Elements | wt % Composition |
| --- | --- |
| C | 0.12–0.18 |
| Si | ≤0.2 |
| Mn | 0.8–1.1 |
| S | ≤0.015 |
| P | ≤0.02 |
| Cr | 1.25–1.5 |
| Mo | 0.8–1.0 |
| V | 0.2–0.3 |
| Fe | Balance |

*2.2. Preparation of 15CDV6 Steel Panels*

Samples of 15CDV6 steel were cut as rectangular panels of dimensions 120 cm × 40 cm × 2 cm. The exposed surface was polished using levigated alumina paste as abrasive in order to obtain mirror surface then degreased with Methyl ethyl ketone (MEK), cleaned by demineralized water, then dried.

*2.3. Epoxy Resin Formulation and Steel Coating*

Two formulations were prepared, one of them had a hardener in addition to the DGEBA (1:2 ratio by weight) (standard epoxy coating) (Figure 1).

**Figure 1.** Molecular structure of the DGEBA and polyaminoamide used in this study.

In the second formulation, with 5 wt % ZP was used (epoxy-ZP coating). The epoxy resin was dissolved in acetone, to it was added the desired amount of zinc phosphate and the produced mixture was stirred at a speed of 1000 rpm for 20 min. The curing agent was then added slowly and the mixture of the three components was stirred at room temperature for 20 min. Produced formulations were applied by an air spray technique on 15CDV6 steel panels. Coated panels were then dried and cured at 60 °C for 1 h. The thickness of coatings was measured to be about 15–25 μm which was measured by a digital coating thickness gauge (Layercheck 750 USB FN, Erichsen Gmbh & Co. KG, Hemer, Germany). In order to get more accuracy for the overall coating thickness, it was measured triply for each sample and mean values are reported.

*2.4. Electrochemical Corrosion Tests and Surafce Characterization of Coating*

Electrochemical evaluation of coatings was carried out by electrochemical impedance spectroscopy (EIS) using a Potentiostat (SP-200, BioLogic, Knoxville, TN, USA) instrument. A three-electrode system was used for the electrochemical test used in this study as is shown in Figure 2. A saline solution with a 3 wt % NaCl concentration was used as a corrosive electrolyte. All electrochemical measurements were performed after 1 h immersion time.

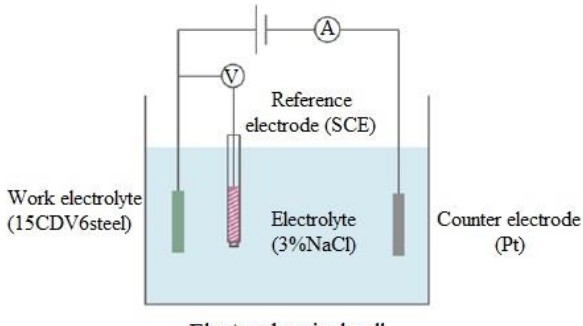

**Figure 2.** Schematic diagram of a three-electrode cell for electrochemical measurement.

In order to get more accuracy and reproducibility of experimental data, the electrochemical studies were performed triply at each tested sample and mean values are reported.

The EIS studies were carried out at a frequency range of 100 kHz to 10 mHz with an alternating current (AC) amplitude of ±10 mV at open circuit potential (OCP). The data points for each frequency were averaged to produce the 37 EIS data points.

Finally, the samples were exposed to the Neutral Salt Spray (NSS) test according to ASTM-B117 [22] before and after 4392 and 5865 h of exposure to the accelerated environment. The surface morphologies of the polymer coatings were characterized by scanning electron microscopy (SEM, S3000H, Hitachi, Tokyo, Japan), with an accelerating voltage of 20 kV.

## 3. Results and Discussion

### 3.1. Fourier-Transform Infrared Spectroscopy (FT-IR) Analysis

The Fourier Transform Infra-Red spectroscopy (FT-IR, FTS 6000, Digilab, Randolph, MA, USA) spectrum of polymeric epoxy cured with polyaminoamide on the 15CDV6 steel surface is shown in Figure 3.

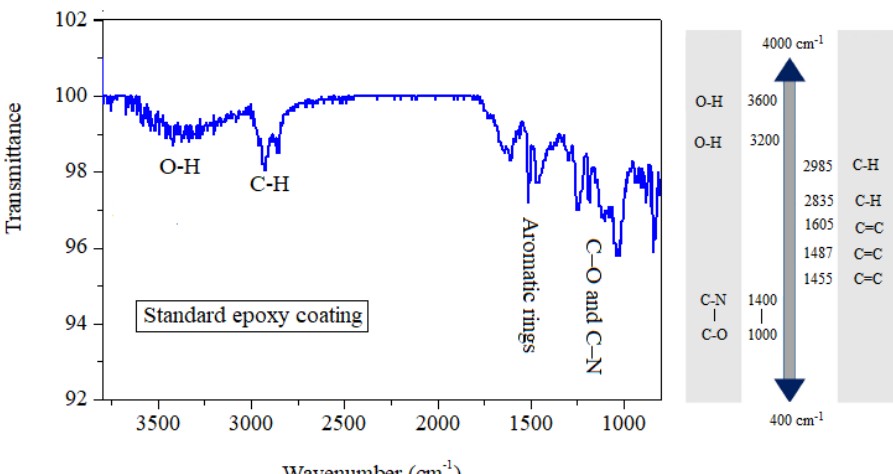

**Figure 3.** The Fourier-transform infrared spectroscopy (FT-IR) spectrum of polymeric epoxy cured with polyaminoamide on the 15CDV6 steel surface.

### 3.2. EIS Measurements

The Bode and Nyquist plots of standard epoxy coating and epoxy-ZP coating after 1464, 2928, and 4392 h exposure are shown in Figures 4 and 5.

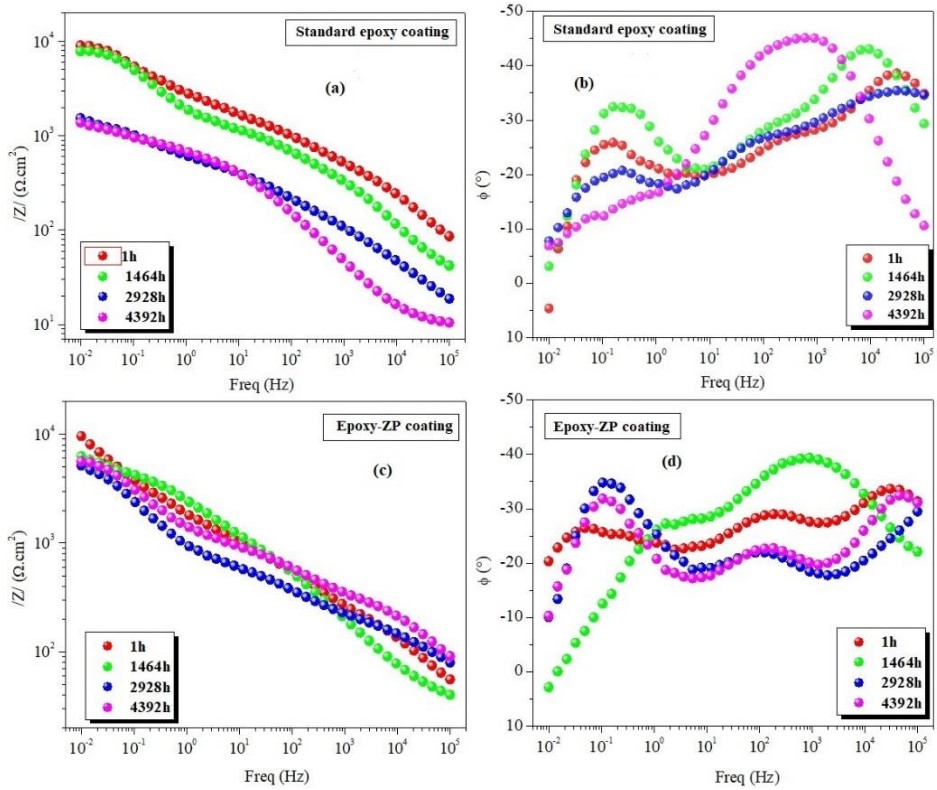

**Figure 4.** Bode magnitude plot (**a**,**c**) and Bode phase (**b**,**d**) plot obtained for the two polymer coatings applied on 15CDV6 steel with an area of 1 cm$^2$ before and after exposure for different times.

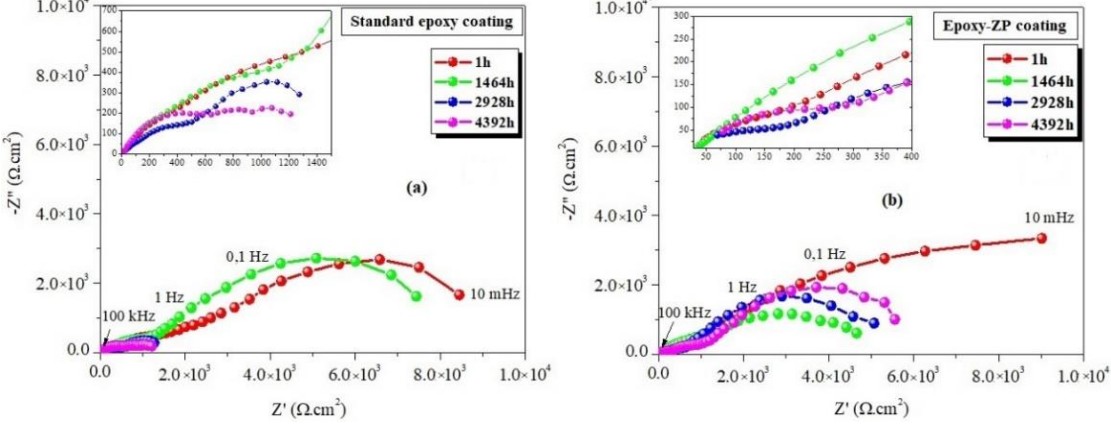

**Figure 5.** Nyquist plots obtained for the two polymer coatings applied on 15CDV6 steel before and after exposure for different times: (**a**) Standard epoxy coating and (**b**) Epoxy-ZP coating.

The Bode plot is a combination of the Bode magnitude plot and Bode phase plot which shows the magnitude of high frequency and low frequency regions due to coating capacitance and charge transfer process, respectively [23]. The impedance modulus (Bode magnitude) at low frequency ($|Z|_{0.01\,Hz}$) was obtained from EIS analysis and is presented in Figure 4. Results of Figure 4 clearly show the impedance diagrams of the standard epoxy coating and epoxy-ZP coating, they are characterized by two time constant.

The impedance modulus at 0.01 Hz was high, greater than 8 kΩ·cm$^2$. As can be seen from Figure 3, the total impedance of the epoxy coating in the high frequency region ($9 \times 10^3$ kHz) after 1 h of immersion is slightly greater than that of the standard epoxy coating. The $|Z|_{0.01\,Hz}$ values of the standard epoxy coating started at 8.90 kΩ·cm$^2$ after 1 h immersion and dropped to 1.40 kΩ·cm$^2$ after

4392 h of exposure. The decrease in the impedance values with immersion time could be related to the dispersion of the corrosive electrolyte into the polymer coating, thus the barrier properties of the film are reduced. Further, the impedance for the epoxy-ZP coating is significantly higher than that for the standard epoxy coating, after 1 h of exposure it was 9.50 k$\Omega$·cm$^2$ and then dropped to 5.90 k$\Omega$·cm$^2$ after 4392 h of exposure. This shows the good barrier properties of epoxy-ZP coating, which could be attributed to the presence of the pigments as well as the epoxy resin. The pigments fill the micro holes and cuts in the epoxy coating, as a result of that the epoxy resin porosity diminishes [24–26].

The Nyquist plots of standard epoxy coating and epoxy-ZP coating after 1464, 2928, and 4392 h exposure are shown in Figure 5.

Bode phase plot ($\theta_{10\,kHz}$) at a high frequency is another parameter that was used to evaluate the protection performance of the polymer coatings. The $\theta_{10\,kHz}$ at a high frequency of standard epoxy coating and epoxy-ZP coating after a 1 h of immersion were about −31° and −36°, respectively. However, the $\theta_{10\,kHz}$ of standard epoxy coating decreases to about −30° after 4392 h of exposure. The drop in the phase angle could be due to the dispersion of the corrosive electrolyte into the polymer coating. Further, the $\theta_{10\,kHz}$ of the epoxy-ZP coating drops to about −33° after 4392 h of exposure. This shows the good barrier properties of epoxy-ZP coating.

The electrical equivalent circuit was used in this study as shown in Figure 6.

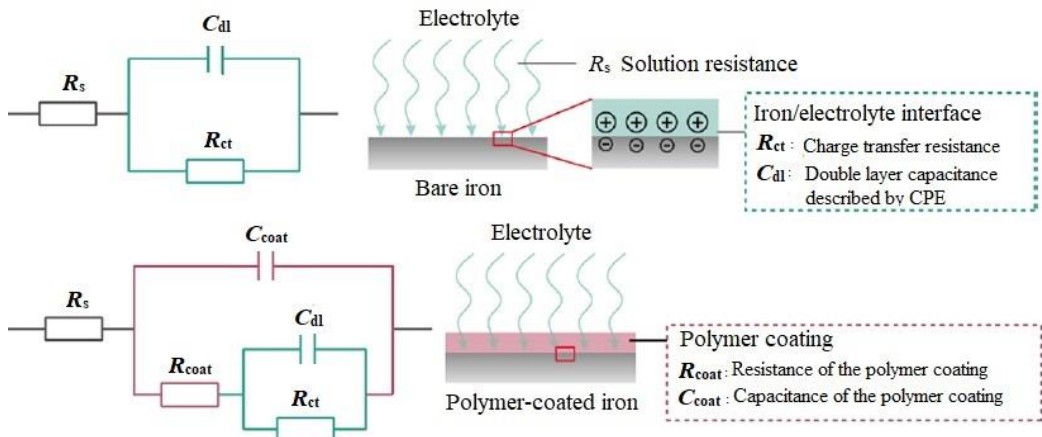

**Figure 6.** The equivalent electrical circuit used.

The Capacitance of the polymer coating values are calculated using Equation (1) [27]:

$$C_x = \frac{(R_x CPE)^{\frac{1}{n}}}{R_x} \tag{1}$$

where constant phase element (*CPE*) is non-ideal capacity and *n* is a capacity factor. The EIS data for the two polymer coatings are given in Table 3.

The results obtained are reported in Table 3, show that the resistance of the polymer coating ($R_{coat}$) and charge transfer resistance ($R_{ct}$) values for epoxy-ZP coating were greater than that for the standard epoxy coating, an indication of the superiority of the epoxy-ZP coating over the standard epoxy coating.

After 1 h of immersion, the $R_{coat}$ and $R_{ct}$ values of the epoxy-ZP coating were much higher than the one of the standard epoxy coating. After the first 1464 h of immersion the $R_{coat}$ and $R_{ct}$ values dropped for both the standard epoxy coating and the epoxy-ZP coating, a higher decrease was seen from the standard epoxy coating. After 2928 h of immersion, the $R_{coat}$ and $R_{ct}$ values of standard epoxy coating continued to decrease, while $R_{coat}$ and $R_{ct}$ values of epoxy-ZP coating actually showed an increase. As mentioned above, the observed results could be attributed to the penetration of the standard epoxy coating by water and electrolytes. While in the epoxy-ZP coating the penetration

was blocked by the zinc phosphate micro-particles. The immersion period between 2928 and 4392 h showed a continuous decrease of in the $R_{coat}$ and $R_{ct}$ values of the standard epoxy coating, while the epoxy-ZP coating showed a continuous increase in the $R_{coat}$ and $R_{ct}$ values. Extending the immersion time to long period caused the water and corrosive electrolytes to reach the coating–metal interface causing an oxidation of active sites on the metal surface, leading to the formation of corrosion on the steel surface. While with the epoxy-ZP coating, the presence of ZP significantly improved the barrier properties of the polymer coating as mentioned earlier.

**Table 3.** The electrochemical parameters extracted from electrochemical impedance spectroscopy (EIS) data for two polymer coatings applied on 15CDV6 steel samples immersed in corrosive media for 1 h at room temperature.

| Sample | Time (h) | Magnitude $\lvert Z\rvert_{0.01\ Hz}$ ($k\Omega\cdot cm^2$) | Phase $-\theta_{10\ kHz}$ (deg) | $R_s$ ($\Omega\cdot cm^2$) | CPE$_{coat}$ $C_{coat}$ ($\mu F/cm^2$) | $n_{coat}$ | $R_{coat}$ ($k\Omega\cdot cm^2$) | CPE$_{dl}$ $C_{dl}$ ($\mu F/cm^2$) | $n_{dl}$ | $R_{ct}$ ($k\Omega\cdot cm^2$) | $\chi^2$ |
|---|---|---|---|---|---|---|---|---|---|---|---|
| Standard epoxy coating | 1 | 8.90 | 31 | 25.01 | 65.6 | 0.36 | 6.90 | 15.0 | 0.74 | 6.62 | 0.21 |
| | 1464 | 7.64 | 33 | 11.54 | 30.3 | 0.51 | 2.13 | 20.0 | 0.73 | 6.60 | 0.18 |
| | 2928 | 1.60 | 33 | 03.39 | 35.1 | 0.36 | 0.73 | 10.4 | 0.66 | 1.07 | 0.03 |
| | 4392 | 1.40 | 30 | 09.37 | 48.8 | 0.68 | 0.03 | 51.2 | 0.19 | 2.00 | 0.07 |
| Epoxy-ZP coating | 1 | 9.50 | 36 | 16.7 | 24.0 | 0.30 | 6.16 | 89.2 | 1.00 | 16.60 | 0.11 |
| | 1464 | 6.65 | 43 | 24.6 | 53.3 | 0.52 | 3.42 | 10.4 | 0.66 | 4.21 | 0.11 |
| | 2928 | 5.10 | 36 | 6.11 | 33.9 | 0.26 | 2.25 | 19.3 | 0.77 | 7.53 | 0.09 |
| | 4392 | 5.90 | 33 | 68.9 | 29.7 | 0.22 | 2.50 | 10.0 | 1.00 | 8.50 | 0.17 |
| | 5136 | 7.60 | 38 | 45.8 | 10.1 | 0.62 | 2.93 | 38.6 | 0.33 | 13.6 | 0.52 |
| | 5496 | 6.90 | 33 | 51.0 | 9.55 | 0.64 | 2.89 | 51.4 | 0.38 | 6.46 | 0.82 |
| | 5856 | 2.30 | 27 | 12.7 | 22.0 | 0.51 | 1.93 | 20.5 | 0.48 | 7.94 | 0.67 |

The total impedance values extracted from the Nyquist and Bode plots were used to assess the stability of the coating during the exposure to the electrolytes for a long period of time [28]. It was also used to monitor the phenomena occurring on the coating–metal interface during the immersion of the coated steel in an electrolyte solution at various periods of time and for a long immersion time (4392 h).

The Bode (magnitude and phase) and Nyquist plots for the epoxy-ZP coating after exposure for different times are shown in Figure 7a–c.

The obtained electrochemical parameters are given in Table 3. The $\lvert Z\rvert_{0.01\ Hz}$ values of the epoxy-ZP coating starts at 7.60 $k\Omega\cdot cm^2$ after 5136 h of exposure and reaches 2.30 $k\Omega\cdot cm^2$ after 5856 h of exposure. On the other hand, the Bode phase plots $\theta_{10\ kHz}$ values of the epoxy-ZP coating begins at $-38°$ after 5136 h immersion and reaches $-27°$ after 5856 h of exposure. As shown in Table 3, the $R_{coat}$ values of the epoxy-ZP coating showed a slight drop in the $R_{coat}$ value from 2.93 to 1.93 $k\Omega\cdot cm^2$ after 5136 and 5856 h immersion, respectively. The slight drop in the $R_{coat}$ values of the epoxy-ZP coating demonstrates that this coating has unique physical properties that allowed it to control the penetration of the electrolytes to the metal surface even after a long period of exposure to the corrosive electrolytes.

The corrosion rate is described by the charge transfer resistance ($R_{ct}$). For the epoxy-ZP coating it is observed that the value of $R_{ct}$ was 13.6 $k\Omega\cdot cm^2$ after 5136 h of immersion in a corrosive electrolyte, while it is 6.46 and 6.01 $k\Omega\cdot cm^2$ after 5496 and 5856 h immersion, respectively.

In brief, zinc phosphate works by inhibiting the formation of a passivation layer containing Zn/P on the 15CDV6 steel, as schematically summarized in Figure 8.

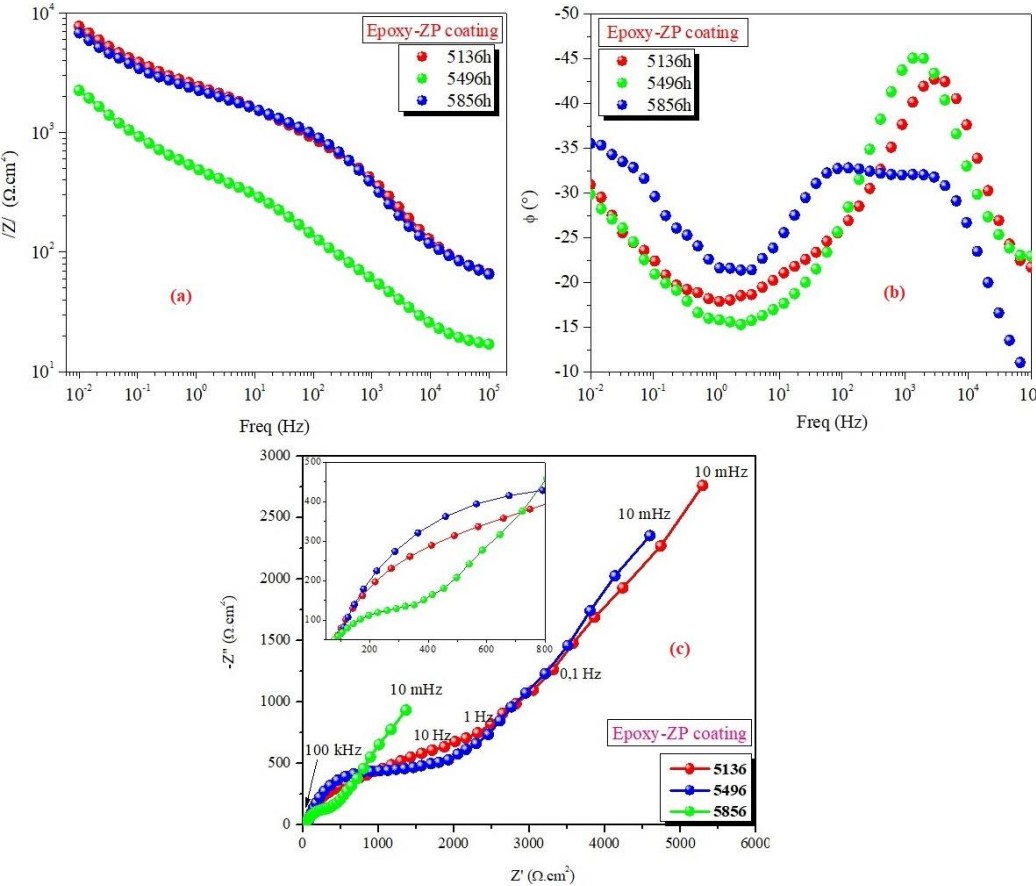

**Figure 7.** Bode (magnitude (**a**) and phase (**b**)) and Nyquist plots (**c**) obtained for the epoxy-ZP coating applied on 15CDV6 steel samples before and after exposure for different times.

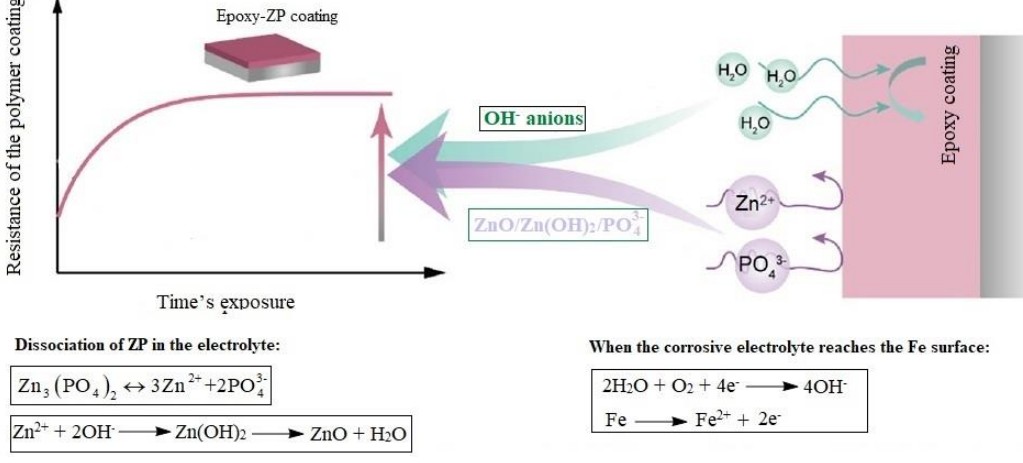

**Figure 8.** Schematic proposed for epoxy-ZP coating applied on 15CDV6 steel samples after exposure.

### 3.3. Surface Morphology of the Coatings

Scanning electron microscopy analysis was performed to investigate the surface morphology of the two evaluated polymer coatings: Standard epoxy coating and the epoxy-ZP coating. The coating surface exposed to an electrolyte solution in a spray test chamber was also evaluated at various times of exposure. Results are shown in Figures 9 and 10.

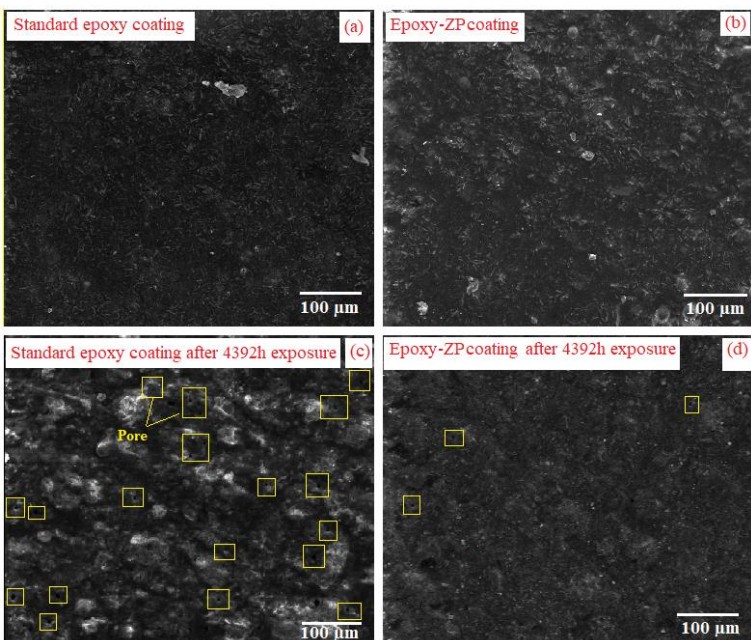

**Figure 9.** SEM-images of the standard epoxy coating (**a**) and epoxy-ZP coating (**b**) applied on 15CDV6 steel before and after 4392 h exposure (**c,d**, respectively).

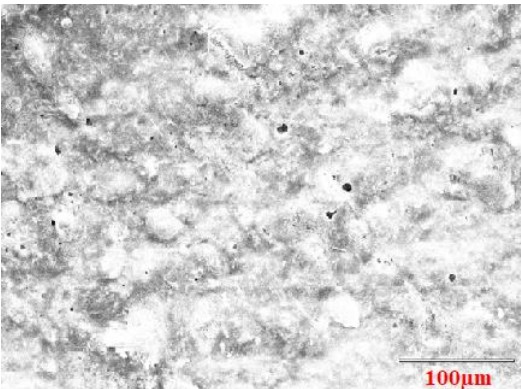

**Figure 10.** SEM image for 15CDV6 steel coated with epoxy-ZP coating after 5856 h of exposure.

As shown in Figure 9a, before the exposure, the standard epoxy coating surface is uniform, smooth and free from defects with a low number of aggregates. The polymer coatings containing 5 wt % zinc phosphate (Figure 9b), shows that zinc phosphate is well dispersed in the coating layer, and is more uniform and homogeneous than the standard coating.

After 4392 h of exposure, the standard epoxy coating (Figure 9c) showed porous morphology and the corrosion site can be observed on the 15CDV6 steel surface. However, the surface of the epoxy-ZP coating showed fewer scratches and corrosion sites Figure 9d. As mentioned above, the results confirm that the addition of zinc phosphate facilitated the formation of a passive phosphate barrier film on the 15CDV6 steel substrate and inhibited the penetration of corrosive electrolyte ions ($H_2O$, $O_2$, and $Cl^-$) to the 15CDV6 steel. It also enhances the anticorrosion process after 4392 h exposure.

The surface morphology of the epoxy-ZP coating was also evaluated after 5856 h of exposure. The obtained SEM micrograph is presented in Figure 10. From Figure 10 it can be obviously seen that the surface morphology of epoxy coating containing ZP is significantly changed. The surface became more porous and the epoxy-ZP coating surface can be seen in the figure. These observations show that surface corrosion sites also developed with the epoxy coating containing ZP.

## 4. Conclusions

In this study, two polymer-based coatings (standard epoxy coating) and (epoxy-ZP coating) were evaluated as an anticorrosive for steel 15CDV6. The studies were performed on coated samples before and after exposure. The experimental results represented by EIS results indicate that the epoxy-ZP coating exhibited excellent physical barrier properties and high pore resistance value. So, the dual component coating of the present work is a nontoxic, ecofriendly coating with a high potential in marine corrosion protection. A surface study of the polymer coating by SEM showed a homogeneous distribution of zinc phosphate in the polymer coating.

**Author Contributions:** Methodology, O.D. and S.J.; Results Discussion, B.K. and G.H.; Validation and Supervision, A.E.H.; Writing—Review and Editing, S.J. and O.D; Editing and Financial Support, A.D.

**Funding:** This research received no external funding.

**Acknowledgments:** The authors would like to thank An-Najah National University and Ibn Tofail University for providing the necessary equipment and chemicals for getting this work accomplished. The authors would also like to thank the laboratory of metallurgical analysis, Cetim Maroc Développement and quality control laboratory, Casablanca Aeronautics Group Figeac Aero. Aeronautical Technopole of Nouaceur, Mohammed V-Casablanca Airport, Morocco. Finally, we would like to thank the Palestine Ministry of Higher Education and Research Scientific

**Conflicts of Interest:** The authors declare no conflict of interest.

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
