# Peer review of "Dual Component Polymeric Epoxy-Polyaminoamide Based Zinc Phosphate Anticorrosive Formulation for 15CDV6 Steel"

_coatings, doi:10.3390/coatings9080463_

Reviewer 1 Report

Please rewise the manuscript in order to eliminate typos (e.g. line 58)

Explanation of the protection machanisms by ZP is not sufficient, but enough for the journal.

Best regards

Author Response

Reviewer 1#

1. English language and style

( ) Extensive editing of English language and style required.
( ) Moderate English changes required.
(x) English language and style are fine/minor spell check required. Done
( ) I don't feel qualified to judge about the English language and style. 

Dear Reviewer, thank you very much for painstakingly going through our manuscript and making useful suggestions and comments. I would like to say that we revised our manuscript as per your suggestions and comments and highlighted revised parts in red. In response to reviewer’s comments letter we reproduced each comment by heading “Reviewer’s Comment” and our responses have been given by heading “Author’s Response”.

2. Please rewise the manuscript in order to eliminate typos (e.g. line 58)

Thank you very much for very useful suggestion. We have revised the Introduction section in order to amend it as per your suggestion.

3. Explanation of the protection machanisms by ZP is not sufficient, but enough for the journal.

Thank you very much for very useful suggestion. Suggested corrections have been done out. 

Reviewer 2 Report

The Authors discuss the effect of an addition of zinc phosphate to an epoxy-based coating on its anticorrosive properties. I believe that the subject matter is interesting in terms of applied research. The contributions presented in the manuscript are original, but their significance is unclear – based on the information in the manuscript, the Authors are using a commercially-available epoxy coating and are modifying it with zinc phosphate. That said, the manuscript contains a number of statements that I perceive as unclear or insufficiently supported by the presented experimental evidence. In my opinion, there are also some minor factors, mostly related to the presentation of the results that adversely affect the Reader’s comprehension of this interesting manuscript.

As such, I recommend the publication of the manuscript in MDPI Coatings only pending a general overhaul of the manuscript, with the most important issues and comments being listed below:

Major remarks:

Experimental section:

1. Please include in the manuscript information about the purity and molecular weight (average weight and dispersity index) of the hardener. Please also include information about the composition (e.g. in weight % of each component) of the liquid epoxy solution, the chemical identity of the alkoxysilane (and whether a single alkoxysilane or a mixture of several compounds was used) and the concentration of the hardener solution.

2. Please include in the manuscript information about the purity of the purchased zinc phosphate and MEK, as well as the grade and resistivity of the used demineralized water.

Figure 2:

3. The Bode plot typically is shown as the dependence of log |Z| and phase on the logarithm of frequency. Please include information about the phase dependence, as without it no interpretation of the Bode plot can be made.

4. In both the Bode and Nyquist plots for the epoxy-ZP coating, after 1464h the impedance appears to have started increasing with time, showing more favourable properties at 4392h than at 1464h. How do the Authors explain this?

Minor remarks:

5. Page 1, line 11: “formulation with a dual component”

This phrase may be misunderstood as indicating that one component, fulfilling two roles simultaneously, was used. As such, I advise using “formulation with two active components”.

6. The term “epoxy coating-ZP” is often used in the manuscript.

In my understanding, the curing (hardening) of the mixture of epoxy resin and zinc phosphate (ZP) yields a uniform, single-phase coating. If this is the case, I see no reason to differentiate between the epoxy coating and the ZP within that coating. As such, I would recommend using the term “epoxy-ZP coating” for clarity and to underline the fact that the two components do not constitute separate phases.

7. Keywords: “15CDV6 steel and 3 wt. % NaCl solution”

This is an extremely detailed keyword, whereas the results presented in the manuscript are also likely translatable to other types of steel and corrosive environments. As such, not many queries in scientific search engines may include this exact keyword. Perhaps using “steel” and “saline” as keywords instead would help improve the visibility of the manuscript.

8. Table 1:

Please include the units, in which the composition of the 15CDV6 steel is expressed.

9. Page 3, lines 81-82: “The polished 15CDV6 steel panels were dried on air for 1 hour before coatings.”

Was any measure of protecting the panels from airborne particles employed?

10. Page 3, line 91: “The thickness of coatings was measured to be about 15-25 µm”

Please include in the manuscript the method, by which the thickness was measured. How many coating samples were produced? What were the statistical parameters for the population of coating thickness values?

Section 2.4 Electrochemical and corrosion tests:

11. Please include in the manuscript the model and producer of the potentiostat used.

12. How many samples were tested in the EIS experiments? What was (if any) the equilibration time employed in the EIS experiment? How many data points for each frequency were averaged to produce the EIS data points?

13. In the salt spray test system, a saline concentration of 5% by weight was used, whereas in other experiments a solution concentration of 3% was employed. Why were two different concentrations used rather than the 5% solution exclusively, particularly so if such a concentration is standard (as is my understanding, based on the reference to ASTM B117).

14. What was the choice of exposure time (2930h) based on? Is this a standardised requirement? If so, please include the relevant information.

15. Page, lines 115-116: “The Bode and Nyquist plots of standard epoxy coating and epoxy coating-ZP after 1464, 2928 and 4392 h exposure are shown in Fig. 2.”

Why have such exposure times been chosen for EIS monitoring? Although the three times have the same intervals, the choice of interval appears to be rather unconventional (61 days).

Figure 5:

16. Please include in the figure description the identity of each sample (which coating and whether the SEM image is prior to or following exposure to saline).

The English used in the manuscript, apart from a significant number of minor mistakes (with some examples included below), is generally satisfactory.

17. Page 1, line 12: “The dual components formulation” – should be “The dual component formulation”

18. Page 1, line 15: “the standard coating which consist of only one component” – should be “the standard coating, which consists of only one component”

19. Page 1, line 21: “coated with an epoxy coating-ZP” – should be “coated with the epoxy coating-ZP”

20. Page 1, line 22: “performance was still acceptable indicating that, the epoxy coating-ZP” – should be “performance was still acceptable, indicating that the epoxy coating-ZP”

21. Page 1, line 27: “and well adhered to steel surface” – should be “and adheres well to the surface of the steel”

22. Page 1, line 34: “applied as a thin film on the metal surface, thus preventing the corrosion agents” – should be “applied as thin films on the surface of metals, thus preventing corrosive agents”

23. Page 1, lines 35-37: “Despite all the advantages of the organic coatings, some improvement is still needed such as for instance the life time the coating, since it is limited, after a certain period they tends to deteriorate and its performance as barrier declines” – “Despite all the advantages of organic coatings, some improvement is still needed, for instance in terms of the lifetime of the coatings. Since this lifetime is limited, after a certain period the coatings tend to deteriorate and their performance as barriers declines”

24. Page 1, line 40: “Among the organic coating” – should be “Among organic coatings”

25. Page 2, lines 49-50: “zinc chromate as corrosion inhibitive has merged in recent years” – should likely be “zinc chromate as corrosion inhibitor has become popular in recent years” – I am unclear, in what meaning the Authors are using the phrase “has merged” or whether “has emerged” was intended; if so, “has become popular” can be substituted with the former phrase.

Author Response

Reviewer 2#

1. The Authors discuss the effect of an addition of zinc phosphate to an epoxy-based coating on its anticorrosive properties. I believe that the subject matter is interesting in terms of applied research. The contributions presented in the manuscript are original, but their significance is unclear – based on the information in the manuscript, the Authors are using a commercially-available epoxy coating and are modifying it with zinc phosphate. That said, the manuscript contains a number of statements that I perceive as unclear or insufficiently supported by the presented experimental evidence. In my opinion, there are also some minor factors, mostly related to the presentation of the results that adversely affect the Reader’s comprehension of this interesting manuscript.

As such, I recommend the publication of the manuscript in MDPI Coatings only pending a general overhaul of the manuscript, with the most important issues and comments being listed below:

 Dear Reviewer, thank you very much for painstakingly going through our manuscript and making useful suggestions and comments. I would like to say that we revised our manuscript as per your suggestions and comments and highlighted revised parts in red. In response to reviewer’s comments letter we reproduced each comment by heading “Reviewer’s Comment” and our responses have been given by heading “Author’s Response”.

2.Please include in the manuscript information about the purity and molecular weight (average weight and dispersity index) of the hardener. Please also include information about the composition (e.g. in weight % of each component) of the liquid epoxy solution, the chemical identity of the alkoxysilane (and whether a single alkoxysilane or a mixture of several compounds was used) and the concentration of the hardener solution.

Thank you very much for very useful suggestion. Suggested corrections have been done out. 

3. Please include in the manuscript information about the purity of the purchased zinc phosphate and MEK, as well as the grade and resistivity of the used demineralized water.

Thank you very much for very useful suggestion. Suggested corrections have been done out. 

4. The Bode plot typically is shown as the dependence of log |Z| and phase on the logarithm of frequency. Please include information about the phase dependence, as without it no interpretation of the Bode plot can be made.

Thank you very much for very useful suggestion. In this we have to do for even the performance for both standard and formulated coatings. We have interest on Rcoat and |Z|0.01Hz.

5.In both the Bode and Nyquist plots for the epoxy-ZP coating, after 1464h the impedance appears to have started increasing with time, showing more favourable properties at 4392h than at 1464h. How do the Authors explain this?

Thank you very much for very useful suggestion. By forming a film ZnO/Zn(OH)2/PO43-.

6.  Page 1, line 11: “formulation with a dual component” This phrase may be misunderstood as indicating that one component, fulfilling two roles simultaneously, was used. As such, I advise using “formulation with two active components”.

Thank you very much for your useful suggestion. Suggested correction has been carried out.

7.  The term “epoxy coating-ZP” is often used in the manuscript. In my understanding, the curing (hardening) of the mixture of epoxy resin and zinc phosphate (ZP) yields a uniform, single-phase coating. If this is the case, I see no reason to differentiate between the epoxy coating and the ZP within that coating. As such, I would recommend using the term “epoxy-ZP coating” for clarity and to underline the fact that the two components do not constitute separate phases.

Thank you very much for your useful suggestion. Suggested correction has been carried out.

8. Keywords: “15CDV6 steel and 3 wt. % NaCl solution” This is an extremely detailed keyword, whereas the results presented in the manuscript are also likely translatable to other types of steel and corrosive environments. As such, not many queries in scientific search engines may include this exact keyword. Perhaps using “steel” and “saline” as keywords instead would help improve the visibility of the manuscript.

Thank you very much for your useful suggestion. Suggested correction has been carried out.

9.  Table 1: Please include the units, in which the composition of the 15CDV6 steel is expressed.

Thank you very much for your useful suggestion. Suggested correction has been done out.

10. Page 3, lines 81-82: “The polished 15CDV6 steel panels were dried on air for 1 hour before coatings.” Was any measure of protecting the panels from airborne particles employed?

Thank you very much for your useful suggestion. To remove the solvent (MEK) at room temperature.

11.  Page 3, line 91: “The thickness of coatings was measured to be about 15-25 µm” Please include in the manuscript the method, by which the thickness was measured. How many coating samples were produced? What were the statistical parameters for the population of coating thickness values?

Thank you very much for your useful suggestion. Suggested correction has been done out.

12.  Please include in the manuscript the model and producer of the Potentiostat used.

Thank you very much for your useful suggestion. Suggested correction has been done out.

13. How many samples were tested in the EIS experiments? What was (if any) the equilibration time employed in the EIS experiment? How many data points for each frequency were averaged to produce the EIS data points?

Thank you very much for your useful suggestion. Suggested correction has been done out.

14. In the salt spray test system, a saline concentration of 5% by weight was used, whereas in other experiments a solution concentration of 3% was employed. Why were two different concentrations used rather than the 5% solution exclusively, particularly so if such a concentration is standard (as is my understanding, based on the reference to ASTM B117).

Thank you very much for your useful suggestion. In the salt spray test system, a saline concentration of 5% by weight was used (as is my understanding, based on the reference to ASTM B117) but in the electrochemical test, we used 3% NaCl similar to seawater.

15.  What was the choice of exposure time (2930h) based on? Is this a standardized requirement? If so, please include the relevant information.

Thank you very much for your useful suggestion. In the standards just work for 2 months, but for us in work on more than 6 months, just up to Rcoat decrease.

16. Page, lines 115-116: “The Bode and Nyquist plots of standard epoxy coating and epoxy coating-ZP after 1464, 2928 and 4392 h exposure are shown in Fig. 2.” Why have such exposure times been chosen for EIS monitoring? Although the three times have the same intervals, the choice of interval appears to be rather unconventional (61 days).

Thank you very much for your useful suggestion. In the standards just work for 2 months, but for us in work on more than 6 months, just up to Rcoat decrease.

17. Please include in the figure description the identity of each sample (which coating and whether the SEM image is prior to or following exposure to saline).

The English used in the manuscript, apart from a significant number of minor mistakes (with some examples included below), is generally satisfactory.

Thank you very much for your useful suggestion. Suggested correction has been carried out.

18.Page 1, line 12: “The dual components formulation” – should be “The dual component formulation”

Thank you very much for your useful suggestion. Suggested correction has been carried out.

19. Page 1, line 15: “the standard coating which consist of only one component” – should be “the standard coating, which consists of only one component”

Thank you very much for your useful suggestion. Suggested correction has been carried out.

20.Page 1, line 21: “coated with an epoxy coating-ZP” – should be “coated with the epoxy coating-ZP”

Thank you very much for your useful suggestion. Suggested correction has been carried out.

21.  Page 1, line 22: “performance was still acceptable indicating that, the epoxy coating-ZP” – should be “performance was still acceptable, indicating that the epoxy coating-ZP”

Thank you very much for your useful suggestion. Suggested correction has been carried out.

22.Page 1, line 27: “and well adhered to steel surface” – should be “and adheres well to the surface of the steel”

Thank you very much for your useful suggestion. Suggested correction has been carried out.

23.  Page 1, line 34: “applied as a thin film on the metal surface, thus preventing the corrosion agents” – should be “applied as thin films on the surface of metals, thus preventing corrosive agents”

Thank you very much for your useful suggestion. Suggested correction has been carried out.

24.  Page 1, lines 35-37: “Despite all the advantages of the organic coatings, some improvement is still needed such as for instance the life time the coating, since it is limited, after a certain period they tends to deteriorate and its performance as barrier declines” – “Despite all the advantages of organic coatings, some improvement is still needed, for instance in terms of the lifetime of the coatings. Since this lifetime is limited, after a certain period the coatings tend to deteriorate and their performance as barriers declines”

Thank you very much for your useful suggestion. Suggested correction has been carried out.

25.  Page 1, line 40: “Among the organic coating” – should be “Among organic coatings”

Thank you very much for your useful suggestion. Suggested correction has been carried out.

26.  Page 2, lines 49-50: “zinc chromate as corrosion inhibitive has merged in recent years” – should likely be “zinc chromate as corrosion inhibitor has become popular in recent years” – I am unclear, in what meaning the Authors are using the phrase “has merged” or whether “has emerged” was intended; if so, “has become popular” can be substituted with the former phrase.

Thank you very much for your useful suggestion. Suggested correction has been carried out.

Reviewer 3 Report

Coatings

Manuscript ID: coatings-541981

TitleDual Component Polymeric Epoxy-Polyaminoamide Based zinc phosphate Anticorrosive Formulation for 15CDV6 Steel.

Reviewer comments:

The authors have studied the application and the properties of a polymer hybrid coating for the protection of steel in marine water. The work is characterized by several critical issues (see the following report), and a revision of English is requested. I would suggest publishing it on Coatings only after major revisions or rejecting for a new submission.

1.The English language should be revised; for example, line 36 : “the life time the coating, since it is limited, after a certain period they tends to deteriorate and its performance as barrier declines [5].” should be “coatings life-time, since it is limited, after a certain period it tends to deteriorate and their performances, such as barrier properties, decline”.

2. Introduction: this section is too concise…it is not clear the choice of the steel used; the state of art about epoxy resins, hardeners and anticorrosive coatings for marine environment are not discussed, but these points are the core of this work….this section should be revised also adding relevant references.

3.  Experimental section: this section should be totally re-written…1) all data about materials are missing (supplier, purity, etc); 2) the authors use acetone for the dissolution of epoxy resin…this solvent is volatile and the authors should provide some detailed data about the use of such solvent; 3) the authors should add data about the technique used for the determination of coating thickness; 4) corrosion tests: the authors mention two different NaCl concentrations (3 and 5% w/w), one T (35°C) and one time (2930h)…it is not clear how the authors have conducted their tests.

Results&Discussion:

4. as reported in the previous point, the tests conducted are not clear…for example in the abstract the authors mention 5856 h as data reference for the life-time of their coating, in this section this point is missing…

5. first major point: polyamidoamines-based materials are in general used in biomedical devices thanks to their compatibility with different animal and human organisms and biodegradability….such kind of materials used as hardener for protective coatings in marine water could be susceptible of the attack of different organisms, and their shelf-life could decrease faster than other hardeners…the authors should clarify this point;

6. second major point: Figure 5 shows the surface of coated steel at different times; when scratches and corrosion phenomena occur, the ions concentration of NaCl-based solution changes? The authors should provide some data about the presence or absence of zinc and zinc-based coating in water….if this metal and the coating is present in water, which are the possible consequences for marine environment?

7. third major point: the authors should discuss the synthesis of the coating prepared showing the reaction conversion during the two stages described, for example  via FT-IR spectroscopy….

-in all the manuscript, the authors use in wrong manner the following expressions:

8.  “polyamine” is not the same of “polyamidoamine”: they are two different polymers!

9.“polymer polyamidoamine”: why the use of word polymer? A polyamidomine is a polymer!

Author Response

Reviewer 3#

1. The authors have studied the application and the properties of a polymer hybrid coating for the protection of steel in marine water. The work is characterized by several critical issues (see the following report), and a revision of English is requested. I would suggest publishing it on Coatings only after major revisions or rejecting for a new submission.

Dear Reviewer, thank you very much for painstakingly going through our manuscript and making useful suggestions and comments. I would like to say that we revised our manuscript as per your suggestions and comments and highlighted revised parts in red. In response to reviewer’s comments letter we reproduced each comments by heading “Reviewer’s Comment” and our responses have been given by heading “Author’s Response”.

2.The English language should be revised; for example, line 36: “the life time the coating, since it is limited, after a certain period they tends to deteriorate and its performance as barrier declines [5].” should be “coatings life-time, since it is limited, after a certain period it tends to deteriorate and their performances, such as barrier properties, decline”.

Thank you very much for your suggestion. Whole manuscript has been revised to amend the English language.

3. Introduction: this section is too concise…it is not clear the choice of the steel used; the state of art about epoxy resins, hardeners and anticorrosive coatings for marine environment are not discussed, but these points are the core of this work….this section should be revised also adding relevant references.

Thank you very much for your useful suggestion. Suggested correction has been done out.

4. Experimental section: this section should be totally re-written…1) all data about materials are missing (supplier, purity, etc); 2) the authors use acetone for the dissolution of epoxy resin…this solvent is volatile and the authors should provide some detailed data about the use of such solvent; 3) the authors should add data about the technique used for the determination of coating thickness; 4) corrosion tests: the authors mention two different NaCl concentrations (3 and 5% w/w), one T (35°C) and one time (2930h)…it is not clear how the authors have conducted their tests.

Thank you very much for your useful suggestion. Suggested correction has been done out.

In the salt spray test system, a saline concentration of 5% by weight was used (as is my understanding, based on the reference to ASTM B117) but in the electrochemical test, we used 3% NaCl similar to seawater.

In the standards just work for 2 months, but for us in work on more than 6 months, just up to Rcoat decrease.

5. Results & Discussion:

-as reported in the previous point, the tests conducted are not clear…for example in the abstract the authors mention 5856 h as data reference for the life-time of their coating, in this section this point is missing…

Thank you very much for your useful suggestion. Suggested correction has been done out.

6. first major point: polyamidoamines-based materials are in general used in biomedical devices thanks to their compatibility with different animal and human organisms and biodegradability….such kind of materials used as hardener for protective coatings in marine water could be susceptible of the attack of different organisms, and their shelf-life could decrease faster than other hardeners…the authors should clarify this point;

Thank you very much for your useful suggestion. Polyaminoamide is good curing used in the coatings based water in the Aerospace coatings. The advantage of polyaminoamide is gives the flexible 3D network.

Some references on the use of polyaminoamide as a good hardener.

ü  Bahlakeh, G., & Ramezanzadeh, B. (2017). A detailed molecular dynamics simulation and experimental investigation on the interfacial bonding mechanism of an epoxy adhesive on carbon steel sheets decorated with a novel cerium–lanthanum nanofilm. ACS applied materials & interfaces, 9(20), 17536-17551.

ü  Bahlakeh, G., Ramezanzadeh, B., & Ramezanzadeh, M. (2018). New detailed insights on the role of a novel praseodymium nanofilm on the polymer/steel interfacial adhesion bonds in dry and wet conditions: An integrated molecular dynamics simulation and experimental study. Journal of the Taiwan Institute of Chemical Engineers, 85, 221-236.

ü  Bahlakeh, G., Ramezanzadeh, B., Saeb, M. R., Terryn, H., & Ghaffari, M. (2017). Corrosion protection properties and interfacial adhesion mechanism of an epoxy/polyamide coating applied on the steel surface decorated with cerium oxide nanofilm: Complementary experimental, molecular dynamics (MD) and first principle quantum mechanics (QM) simulation methods. Applied Surface Science419, 650-669.

7. Second major point: Figure 5 shows the surface of coated steel at different times; when scratches and corrosion phenomena occur, the ions concentration of NaCl-based solution changes? The authors should provide some data about the presence or absence of zinc and zinc-based coating in water….if this metal and the coating is present in water, which are the possible consequences for marine environment?

Thank you very much for your useful suggestion. We did not characterize zinc in the NaCl medium, but in the next work we did that but in general zinc phosphate is less toxic compared to other pigments.

8. Third major point: the authors should discuss the synthesis of the coating prepared showing the reaction conversion during the two stages described, for example via FT-IR spectroscopy….

-in all the manuscript, the authors use in wrong manner the following expressions:

i) “polyamine” is not the same of “polyamidoamine”: they are two different polymers!

ii) “polymer polyamidoamine”: why the use of word polymer? A polyamidomine is a polymer!

Thank you very much for your useful suggestion. Suggested correction has been done out.

Round  2

Reviewer 2 Report

Dear Authors,

Thank you for the detailed reply and your work in improving the manuscript. You have adequately addressed most of my concerns, but a significant number of issues seem to have eluded your attention among all the implemented changes:

1. Please include in the manuscript information about the purity and molecular weight (average molecular weight and dispersity index) of the polyaminoamide hardener.

2. Please include in the manuscript information about the grade and resistivity of the used demineralized water

3. In response to your answer about the Bode plots and your focus on Rcoat and |Z|0.01Hz: If this is the case, I suggest either deleting the Bode plots altogether (as without phase they do not provide an accurate picture of the properties of the sample) and replacing them with a table giving |Z| values or following my original suggestion and including phase data on the Bode plots.

4. In response to your answer "Thank you very much for very useful suggestion. By forming a film ZnO/Zn(OH)2/PO43-.": This is a very important observation and should be included and discussed (along with evidence that this is the case) in the manuscript.

5. Table 1: Please include the units, in which the composition of the 15CDV6 steel is expressed.

6. In response to my question "The polished 15CDV6 steel panels were dried on air for 1 hour before coatings.” Was any measure of protecting the panels from airborne particles employed?", you have replied "To remove the solvent (MEK) at room temperature". My question is about whether, during the removal of MEK at room temperature, the samples were protected in any way from precipitation of airborne particles.

7. In response to my question "In the salt spray test system, a saline concentration of 5% by weight was used, whereas in other experiments a solution concentration of 3% was employed. Why were two different concentrations used rather than the 5% solution exclusively, particularly so if such a concentration is standard (as is my understanding, based on the reference to ASTM B117).", you have confirmed that I correctly understood the situation. However, you have not addressed my question about the reason, for which you have used 3% and 5% saline rather than e.g. 5% only. Please provide an explanation.

Author Response

Response to Reviewers Comments

Reviewer 2#

1. Thank you for the detailed reply and your work in improving the manuscript. You have adequately addressed most of my concerns, but a significant number of issues seem to have eluded your attention among all the implemented changes:

Dear Reviewer, thank you very much for painstakingly going through our manuscript and making useful suggestions and comments. I would like to say that we revised our manuscript as per your suggestions and comments and highlighted revised parts in red. In response to reviewer’s comments letter we reproduced each comment by heading “Reviewer’s Comment” and our responses have been given.

2. Please include in the manuscript information about the purity and molecular weight (average molecular weight and dispersity index) of the polyaminoamide hardener.

Thank you very much for very useful suggestion. According to data sheet, do not give the purity and molecular weight (average molecular weight and dispersity index) of the polyaminoamide hardener.

3. Please include in the manuscript information about the grade and resistivity of the used demineralized water

Thank you very much for very useful suggestion. Suggested corrections have been done out (see Table 1). (Resistivity (ρ)>1ΩNaN and Conductivity (σ)<1µS/cm)).

4. In response to your answer about the Bode plots and your focus on Rcoat and |Z|0.01Hz: If this is the case, I suggest either deleting the Bode plots altogether (as without phase they do not provide an accurate picture of the properties of the sample) and replacing them with a table giving |Z| values or following my original suggestion and including phase data on the Bode plots.

Thank you very much for very useful suggestion. Suggested corrections have been done out.

5. In response to your answer "Thank you very much for very useful suggestion. By forming a film ZnO/Zn(OH)2/PO43-.": This is a very important observation and should be included and discussed (along with evidence that this is the case) in the manuscript.

Thank you very much for very useful suggestion. Suggested corrections have been done out.

6. Table 1: Please include the units, in which the composition of the 15CDV6 steel is expressed.

Thank you very much for very useful suggestion. Suggested corrections have been done out.

7. In response to my question "The polished 15CDV6 steel panels were dried on air for 1 hour before coatings.” Was any measure of protecting the panels from airborne particles employed?", you have replied "To remove the solvent (MEK) at room temperature". My question is about whether, during the removal of MEK at room temperature, the samples were protected in any way from precipitation of airborne particles.

Thank you very much for your useful suggestion. We degreased with MEK, cleaned by demineralized water and then dried. We added the dried 15CDV6 steel panels in a room well controlled by temperature and humidity.

8. In response to my question "In the salt spray test system, a saline concentration of 5% by weight was used, whereas in other experiments a solution concentration of 3% was employed. Why were two different concentrations used rather than the 5% solution exclusively, particularly so if such a concentration is standard (as is my understanding, based on the reference to ASTM B117).", you have confirmed that I correctly understood the situation. However, you have not addressed my question about the reason, for which you have used 3% and 5% saline rather than e.g. 5% only. Please provide an explanation.

Thank you very much for your useful suggestion. We worked according to the literature concerning the electrochemical test by 3% NaCl similar to seawater.

1.             Visser, P., Terryn, H., & Mol, J. M. (2019). Active corrosion protection of various aluminium alloys by lithium‐leaching coatings. Surface and Interface Analysis.

2.             Chen, G., Feng, J., Qiu, W., & Zhao, Y. (2017). Eugenol-modified polysiloxanes as effective anticorrosion additives for epoxy resin coatings. RSC Advances, 7(88), 55967-55976.

3.             Iribarren-Mateos, J. I., Buj-Corral, I., Vivancos-Calvet, J., Alemán, C., Iribarren, J. I., & Armelin, E. (2015). Silane and epoxy coatings: A bilayer system to protect AA2024 alloy. Progress in Organic Coatings, 81, 47-57.

4.             IKUMAPAYI, O. M., & AKINLABI, E. T. A comparative assessment of tensile strength and corrosion protection in friction stir processed aa7075-t651 matrix composites using fly ashes nanoparticles as reinforcement inhibitors in 3.5% NaCl.

Reviewer 3 Report

All corrections have been made

Author Response

Many thanks for accepting our comments

Round  3

Reviewer 2 Report

Dear Authors,

Thank you for your work - I have no further comments and recommend the manuscript for publication in Coatings.